# APPLICATION OF DEEP CONVOLUTIONAL NEURAL NETWORK TO PREVENT ATM FRAUD BY FACIAL DISGUISE IDENTIFICATION

## ABSTRACT

The paper proposes and demonstrates a Deep Convolutional Neural Network (DCNN) architecture to identify users with disguised face attempting a fraudulent ATM transaction. The recent introduction of Disguised Face Identification (DFI) framework Singh et al. (2017) proves the applicability of deep neural networks for this very problem. All the ATMs nowadays incorporate a hidden camera in them and capture the footage of their users. However, it is impossible for the police to track down the impersonators with disguised faces from the ATM footage. The proposed deep convolutional neural network is trained to identify, in real time, whether the user in the captured image is trying to cloak his identity or not. The output of the DCNN is then reported to the ATM to take appropriate steps and prevent the swindler from completing the transaction. The network is trained using a dataset of images captured in similar situations as of an ATM. The comparatively low background clutter in the images enables the network to demonstrate high accuracy in feature extraction and classification for all the different disguises.

## 1 INTRODUCTION

The widespread acceptance of Automated Teller Machine (ATM) and their omnipresence in the banking sector has engendered numerous security concerns. One of the most imperative concerns being, verifying the authenticity of the user. Evidently, most of the ATMs across the globe simply rely on a card and a Personal Identification Number (PIN) for authentication. However, in either case, it is plausible that the user is not authorised to transact. For instance, illegal practices like phishing, shoulder surfing, card fraud, stolen card can cause substantial monetary loss to the owner.

To overcome and identify such practices, ATMs have an inbuilt camera which records 24x7. The current state of art ATM security works in the following way: After a fraudulent transaction, the owner of the corresponding bank account reports about the fraud. The police then investigates and goes through the footage recorded by the ATM camera to find the face of the imposter. Once the face is identified, the police searches for the imposter. Clearly, this security measure can be easily gamed by using artifacts or alterations like wigs, caps, eyeglasses, beard to cover the face for intentional disguises. As a result, Righi et al. (2012) stated that such face alterations can substantially degrade the performance of the system. Hence, this approach has a very low success rate which is unacceptable in banking sectors. Additionally, Sharma (2012) explained different openings and vulnerabilities that exist at the time of transactions due to fake entries and fake cards.

Apparently, this chaos can be prevented by ensuring that the transaction proceeds only if the face is undisguised and reveal identity of the user. The proposed system extracts the user's face from the footage and checks if the face is disguised. The system is trained cleverly to identify such faces by an extensive pool of disguised and undisguised faces. If the face is disguised, the system will not allow the transaction to be proceeded, thereby preventing the imposter from stealing.

To achieve this, the proposed system uses Deep Convolutional Neural Networks for image classification using statistical dimensionality reduction method. Deep networks have proved to be exceptional in computer vision problems (Sun et al., 2013)(Haavisto et al., 2013). Sun et al. (2013) stated a three-layer cascading style which superficially captures the high level features and refines them to detect deeper features. Analogously, the proposed system uses a five-layer architecture, first 3

layers comprises of a convolutional layers followed by a pooling layers to learn the features of the following types of images : Disguised, Partially disguised and Undisguised.

## 2 PROPOSED SYSTEM

### 2.1 EXISTING MECHANISMS

Plenty of research work has been published in response to the ATM security problems and a lot of it relates to using machine learning to authenticate users. Fagbolu et al. (2014) proposed a face-based authentication as identity test for users and the system uses facial recognition with biometric features. T. Suganya (2015) stated the applicability of image processing by amalgamation of Face Recognition System (FRS) in the identity verification process engaged in ATMs. Dhamecha et al. (2013) proposed a framework to classify local facial regions of both visible and thermal face images into biometric (regions without disguise) and non-biometric (regions with disguise) classes and used the biometric patches for facial feature extraction and matching.

### 2.2 PROBLEMS & LIMITATIONS

The fact that none of the above mentioned mechanisms are implemented in the current state-of-art endorses that they are not at par with the requirements and have to be compromised due to certain trade-offs. The mechanisms use an extensive pool of user information and glut of computing resources such as cloud storage, network bandwidth which makes them in-feasible and erratic .

According to T. Suganya (2015), transactor's picture should be clicked and matched to existing records after his card and PIN are verified. However, such computation cannot be relied upon as it highly depends on the remote server which holds the data and the available bandwidth. Such systems try to authenticate too much and exceed the computational limits of the machine. Also, emergency situations where the card owner is unavailable to authenticate the transaction is where the current systems (Fagbolu et al., 2014), (T. Suganya, 2015), (Kibona, 2015) suffer. Moreover, with reference to Das & Debbarma (2011), clearly Eigenface based method can be spoofed by using face masks or photos of account holder.

### 2.3 PROPOSED SOLUTION

The proposed solution takes advantage of the fact that fraudsters hide their identity to complete the transaction and not get caught in police investigation. To achieve this, fraudsters cover their faces with disguises while performing the transaction. We propose and demonstrate the use of a Convolutional Neural Network to identify face disguises, thereby preventing fraudulent transactions. An ATM equipped with the proposed DCNN model will not allow any user with disguised face to complete the transaction and prompt for removing the disguise to complete the transaction. The DCNN is trained to identify in real time whether the user interacting with the ATM is having disguised face or not without consuming any additional data or remote resources. The output is swiftly reported to the ATM to further proceed the transaction based on the computed label. The proposed system makes sure that every user interacting with the ATM shows his/her identity to the ATM and essentially allows police to identify the user corresponding to a fraud transaction.

## 3 THE DCNN MODEL

The DCNN model is implemented in python 2.7 using TensorFlow(r1.3) and is executed on an Ubuntu 16.04 operating system. Stochastic Gradient Descent method is used to train the model and Kingma and Ba's Adam algorithm to control the learning rate.

### 3.1 DATASET

The dataset used to train and test the contains 1986 images of faces covered with any combination of 4 disguises namely scarf, helmet, eyeglasses and fake beard. The images are published by Singh et al. (2017) and matches the requirement to an acceptable level. The images are manually classified in 3 classes, which are - disguised, undisguised and partially disguised. Fig. 1 exemplifies the

dataset in short. While feeding to the network the 1986 images were randomly split into 1500 and 486 images which is approximately 75% and 25% split ratio. 1500 images were used for training and remaining 486 for testing. 30 batches of 50 randomly chosen samples were used for every training cycle. After every training cycle, the training set was shuffled to randomize the data and ensure generalization in learning process.

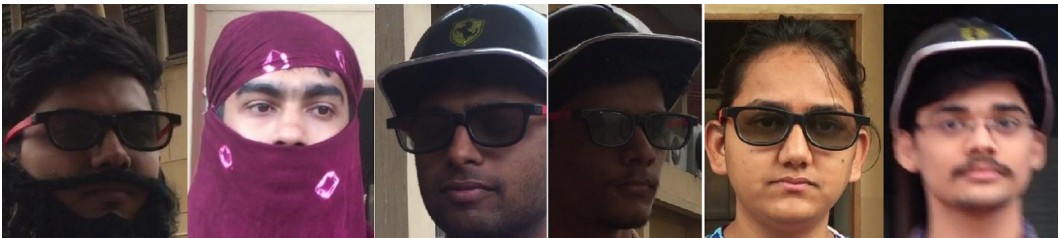

Figure 1: Examples of 2 images of each disguised, partially disguised and undisguised classes respectively.

- **Disguised**

  Faces which are not recognizable by humans are labelled as disguised. These images particularly contain more than one disguises and effectively hide the identity of the person in the image. 1372 images belong to disguised class.

- **Partially Disguised**

  We introduced this label to adopt the network to allow users having unintentional disguises such as spectacles and natural beard. These images are recognizable by humans and the apparent disguises are part of the persons identity. There are 212 samples of partially disguised faces.

- **Undisguised**

  Faces showing clear and effective identity of the person in the image are labelled as undisguised. A total of 402 images belong to the undisguised class.

## 3.2 NETWORK CHARACTERISTICS

The Rectified Linear Unit (ReLU) activation function is used to model the neurons. The following equation defines ReLU function.

$$f(x) = max(0, x) \tag{1}$$

The activation is simply thresholded at zero. Due to its linear and non-saturating form, it significantly accelerates the convergence of Stochastic Gradient Descent (SGD) compared to the sigmoid or tanh functions. The network parameters are initialised with a standard deviation of 0.1 to facilitate SGD.

Furthermore, softmax classification function is used to classify each sample and present the predicted output to cost function to calculate training error. Cross entropy is the loss measure used to measure training error of the learner and it is as defined below

$$C(Y, O) = -\sum_{i=1}^{n} * \log(O_i) \tag{2}$$

where,
n : Number of Samples in the Batch
Y : *[3x1]* Size One-Hot Encoded Vector Showing Correct Classification Label
O : *[3x1]* Size Probability Distribution Vector Obtained from Softmax Function in the Output Layer

The training steps are executed using the **tensorflow.train.AdamOptimizer** function in TensorFlow, with 1e-4 as the initial learning rate and cross entropy as the cost functions needed to be optimized. The AdamOptimizer uses Kingma and Ba's Adam algorithm to control the learning rate. Adam offers several advantages and foremost is that it uses moving averages of the parameters (momentum). Bengio (2012)] discusses the reasons for why this is beneficial. Simply put, this enables Adam to use a larger effective step size, and the algorithm will converge to this step size without fine tuning.

### 3.3 NETWORK ARCHITECTURE AND PARAMETERS

The DCNN model contains 3 convolutional layers each followed by a pooling layer. After the 3rd pooling layer two fully connected hidden layers are introduced to classify the features learned by the convolutional layers. The output layer uses softmax function to calculate probabilities of the image to belong to each class. Figure 2 shows the model architecture along with specific network parameters and respective input and output image dimensions. The first convolutional layer uses 32 feature maps with kernel dimensions as (7x7x3). Similarly, the 2nd and 3rd convolutional layers use 64 and 128 feature maps and kernels with dimensions (5x5x32) and (5x5x64) respectively. All 3 pooling layers use (2x2) max pooling with stride length of 2. The 3 convolutional layers are accompanied with bias vectors of sizes (32x1), (64x1), (128x1) respectively. The following 2 fully connected hidden layers have weight matrices of dimensions (512x18432) and (512x512) respectively and bias vectors of size (512x1) each. The output layer contains 3 neurons and weight matrix of size (3x512) with (3x1) sized bias vector. In all, the network incorporates 9,962,819 trainable parameters which are eventually learned by the network in training step and used to classify the unknown examples while testing.

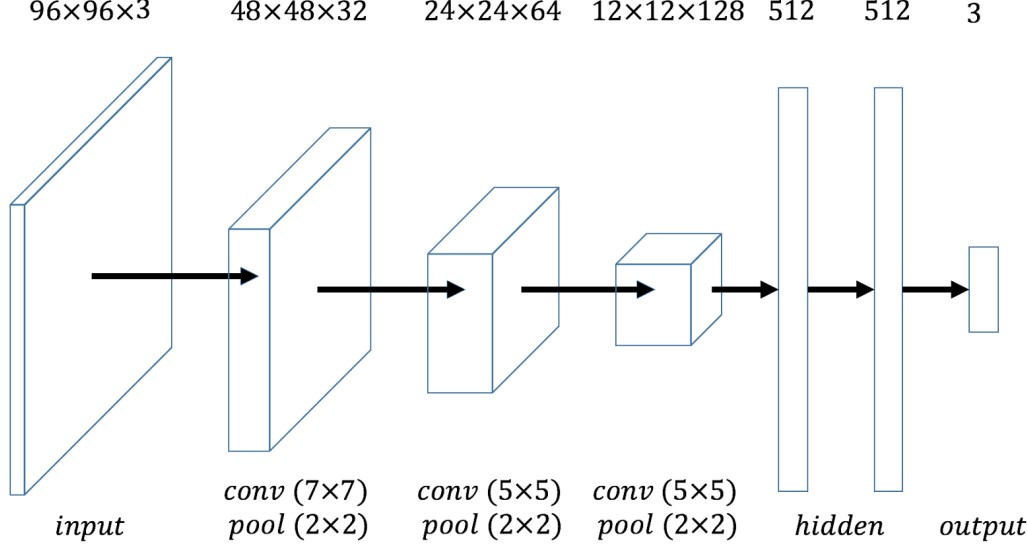

Figure 2: Network architecture and Parameters

## 4 MODEL EXECUTION AND RESULTS

### 4.1 PERFORMANCE

The performance of the model is evaluated on a test set which is disjoint from training set, simulating the real life situation. It reports a final accuracy of 90.535% after around 60 training cycles. Figure 3 shows evolution of test accuracy over 60 training cycles and Figure 4 demonstrates accuracy for every training step for first 5 training cycles.

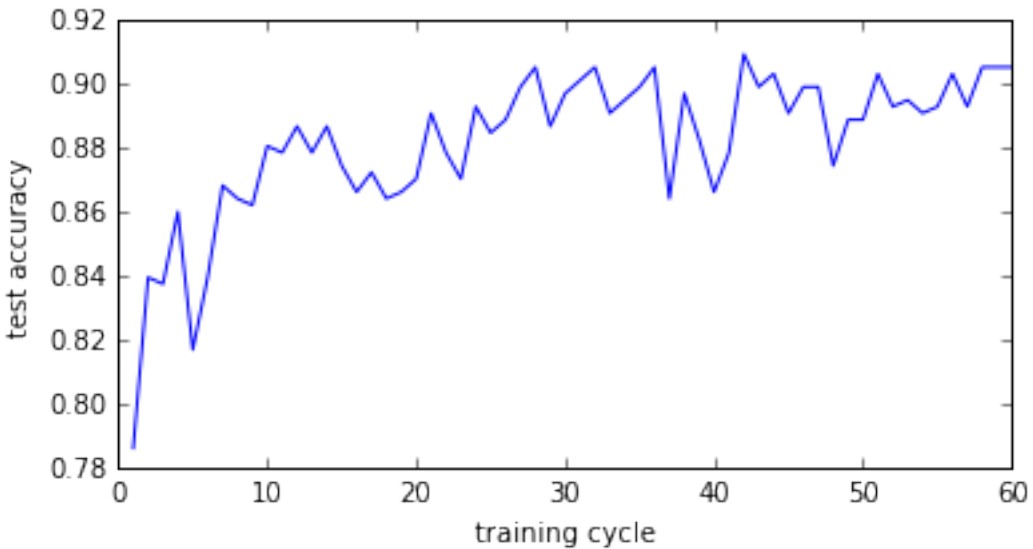

Figure 3: Evolution of test accuracy over 60 training cycles

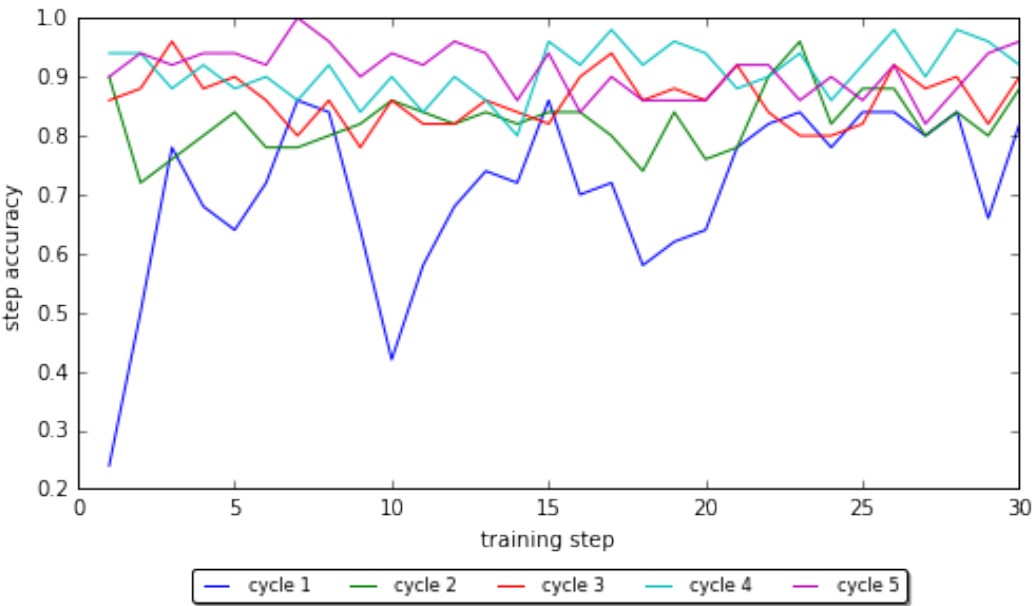

Figure 4: Accuracies for every training step over first 5 cycles

## 4.2 RESULTING ACTIVATIONS

Figures 5-8 show activations generated from the 3 convolutional layers after complete training process. The activations are generated for a disguised face which was correctly classified by the network. They reveal different structures that excite a given feature map, hence showing its invariance to input deformations. The activations in each layer show the hierarchical nature of features learned by the network. The first layer learns corner and border features, second layer points out more deeper features and third layer adds up to learn blur and relatively targeted features for generalization of features. The activations from third convolutional layer are used by fully connected layers for classification.

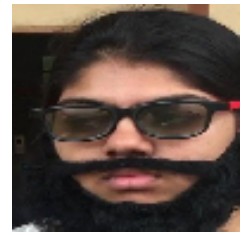

Figure 5: Input image for which activations in Figure 6,7,8 are obtained

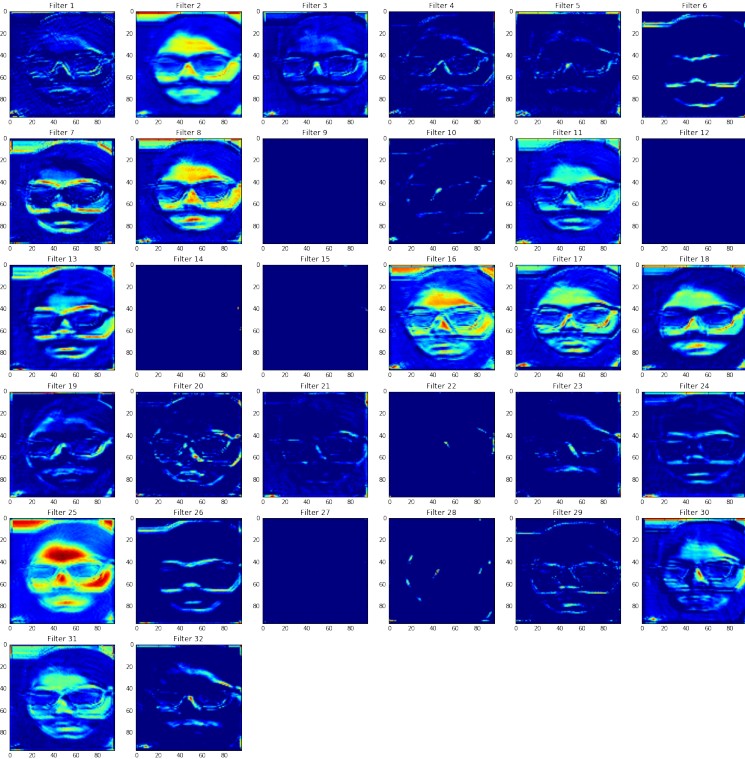

Figure 6: Activations obtained from Conv layer 1 shows the face patches learned by the network

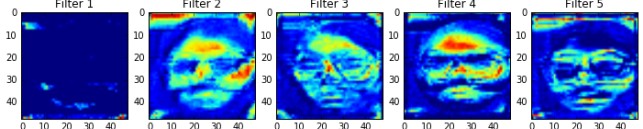

Figure 7: Five activations obtained from filter 1 to 5 in Conv layer 2

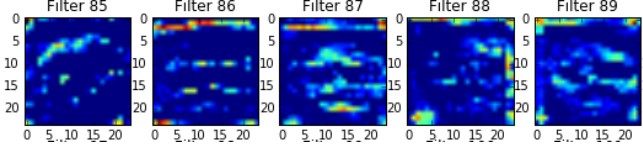

Figure 8: Five activations obtained from filter 85 to 89 in Conv layer 3

### 4.3 System Evaluation and Future Outlook

Neural Networks are proven to work efficiently in numerous classification and regression tasks due to their flexibility and high fault tolerance. The system using Deep Convolutional Neural Network demonstrates high accuracy and works efficiently in real time without needing to access data on cloud. Speed being indispensable to banking transactions is one of the significant advantages of this model. The performance can be significantly improved by training through a large dataset obtained from actual ATMs.

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
