# OpenReview forum: "APPLICATION OF DEEP CONVOLUTIONAL NEURAL NETWORK TO PREVENT ATM FRAUD BY FACIAL DISGUISE IDENTIFICATION"
_ICLR.cc/2018/Conference — Reject_

### Official Review · AnonReviewer1 · 2017-11-22
**clear application paper / class project**

**Rating:** 1
**Confidence:** 5

**Review:**

The paper is relatively clear to follow, and implement.

The main concern is that this looks like a class project rather than a scientific paper. For a class project this could get an A in a ML class!

In particular, the authors take an already existing dataset, design a trivial convolutional neural network, and report results on it. There is absolutely nothing of interest to ICLR except for the fact that now we know that a trivial network is capable of obtaining 90% accuracy on this dataset.

---

### Official Review · AnonReviewer2 · 2017-11-27
**Limited significance and no originality; weak experiments, flaws in the evaluation**

**Rating:** 2
**Confidence:** 4

**Review:**


As one can see by the title, the originality (application of DCNN) and significance (limited to ATM domain) is very limited. If this is still enough for ICLR, the paper could be okay. However, even so one can clearly see that the architecture, the depth, the regularization techniques, and the evaluation are clearly behind the state of the art. Especially for this problem domain, drop-out and data augmentation should be investigated.

Only one dataset is used for the evaluation and it seems to be very limited and small. Moreover, it seems that the same subjects (even if it is other pictures) may appear in the training set and test set as they were randomly selected. Looking into the referece (to get the details of the dataset -  from a workshop of the IEEE International Conference on Computer Vision Workshops (ICCVW) 2017) reveals, that it has only 25 subjects and 10 disguises. This makes it even likely that the same subject with the same disguise appears in the training and test set.

A very bad manner, which unfortunately is often performed by deep learning researchers with limited pattern recognition background, is that the accuracy on the test set is measured for every timestamp and finally the highest accuracy is reported. As such you perform an optimization of the paramerter #iterations on the test set, making it a validation set and not an independent test set.

Minor issues:
make sure that the capitalization in the references is correct (ATM should be capital, e.g., by putting {ATM} - and many more things).

---

### Official Review · AnonReviewer3 · 2017-11-29
**This paper is an application paper on detecting when a face is disguised, however it is poorly written and do not contribute much in terms of novelty of the approach.**

**Rating:** 3
**Confidence:** 5

**Review:**

This paper is an application paper on detecting when a face is disguised, however it is poorly written and do not contribute much in terms of novelty of the approach. The application domain is interesting, however it is simply a classification problem

The paper is written clearly (with mistakes in an equation), however, it does not contribute much in terms of novelty or new ideas.

To make the paper better, more empirical results are needed. In addition, it would be useful to investigate how this particular problem is different than a binary classification problem using CNNs.

Notes:
Equation 2 has a typo, '*'

---

### Decision · Program_Chairs · 2018-01-29
**ICLR 2018 Conference Acceptance Decision**

**Decision:**

Reject

**Comment:**

Reviewers are unanimous that this is a reject.
A "class project" level presentation.
Errors in methodology and presentation.
No author rebuttal or revision